# Gamma Camera Imaging with Rotating Multi-Pinhole Collimator. A Monte Carlo Feasibility Study

**DOI:** 10.3390/s21103367

**Published:** 2021-05-12

**Authors:** Victor Ilisie, Laura Moliner, Constantino Morera, Johan Nuyts, José María Benlloch

**Affiliations:** 1Centro Mixto CSIC, Instituto de Instrumentación para Imagen Molecular (i3M), Universitat Politècnica de València, Camino de Vera, s/n, 46022 Valencia, Spain; lmoliner@i3m.upv.es (L.M.); benlloch@i3m.upv.es (J.M.B.); 2Oncovision, S. A.-Jeronimo de Monsoriu, 92, 46012 Valencia, Spain; cmorera@oncovision.com; 3Faculty of Medicine, KULeuven, B3000 Leuven, Belgium; johan.nuyts@uzleuven.be

**Keywords:** gamma camera, SPECT, PET, mobile collimator, pinhole collimator

## Abstract

In this work, we propose and analyze a new concept of gamma ray imaging that corresponds to a gamma camera with a mobile collimator, which can be used in vivo, during surgical interventions for oncological patients for localizing regions of interest such as tumors or ganglia. The benefits are a much higher sensitivity, better image quality and, consequently, a dose reduction for the patient and medical staff. This novel approach is a practical solution to the overlapping problem which is inherent to multi-pinhole gamma camera imaging and single photon emission computed tomography and which translates into artifacts and/or image truncation in the final reconstructed image. The key concept consists in introducing a relative motion between the collimator and the detector. Moreover, this design could also be incorporated into most commercially available gamma camera devices, without any excessive additional requirements. We use Monte Carlo simulations to assess the feasibility of such a device, analyze three possible designs and compare their sensitivity, resolution and uniformity. We propose a final design of a gamma camera with a high sensitivity ranging from 0.001 to 0.006 cps/Bq, and a high resolution of 0.5–1.0 cm (FWHM), for source-to-detector distances of 4–10 cm. Additionally, this planar gamma camera provides information about the depth of source (with approximate resolution of 1.5 cm) and excellent image uniformity.

## 1. Introduction

Intraoperative surgical imaging is a growing field in molecular imaging [1] and has significantly evolved with the emergence of new techniques such as fluorescence, Raman, photoacoustic and radio-guided techniques, and, lately, deep-learning methods have also been developed and adapted to this purpose [2,3]. It is normally used during surgical interventions for oncological patients for localizing regions of interest such as tumors or ganglia. The localization of sentinel lymph nodes and visual confirmation of the completeness of resection of such nodes are also notable potential application of intraoperative imaging. With this technique, surgical precision (i.e., less extensive surgery) and improved surgical outcomes may result. Currently, there are also new medical challenges, such as in vivo tracking of small number of cells, in the setting of stem cell tissue repair strategies, cancer immunotherapy, time-resolved quantitative multiparametric imaging or pharmacodynamic studies, which call for a paradigm shift in molecular imaging. A great step towards achieving these goals would be enabling one order-of-magnitude leap in the sensitivity and speed of molecular imaging tomographs.

With a sensitivity improvement up to a factor 10 [4,5,6], the proposed idea would permit the reduction of the radiation doses of molecular imaging procedures to negligibly low levels (roughly to a few MBq [7]), the reduction of the synthesized quantity of radiopharmaceutics needed for each examination and, thus, the reduction of the relatively high cost currently associated with in vivo molecular imaging procedures. Although the current application is focused on in vivo surgical monitoring, its benefits can be extended to SPECT and beyond oncology, to cardiovascular, neurological, metabolic, inflammatory, infectious or metabolic disease (such as diabetes), including in the pediatric, neonatal and prenatal contexts, and, potentially, towards molecular in vivo imaging to study the “biology” of the whole human body. Precise dynamic studies of molecular processes are also of high interest in pharmacology for screening and selecting candidate molecules for the next generation of drugs or new applications thereof. By reducing the employed dose by a factor of at least 10, it would also enable the possibility of performing scans more repeatedly in suspect regions, in a safer radiation environment.

The majority of multi-pinhole, multiple-slit or slit-slat collimators are designed to avoid the superposition among the different projections of the gamma ray incidence area, over the surface of the detector. This is due to the fact that this multiplexing ambiguity, i.e., not knowing through which pinhole aperture a particular gamma ray traveled and subsequently struck the detector, produces significant artifacts in the final image. On the other hand, the lack of at least a small percentage of overlapping area translates to an incomplete truncated medical image after reconstruction [8,9,10,11]. Lately, some authors have treated this problem by using multi-pinhole focusing plates, fan-beam, slit-slat and multi-pinholes applied to molecular breast imaging, but with a static configuration [12,13].

During the last decade, interesting studies have provided useful ideas on how to obtain artifact-free images in multiplexed systems. In many systems, it has been observed that it is less probable to generate artifacts with irregular slat shapes or pinhole distributions. This indicates that a high field of view (FOV) sampling is very important, and several groups have successfully combined multiplexing and high sampling [14,15,16]. Another solution consists in obtaining a high number of projections at different distances from the object (synthetic collimation) [17]. Coded aperture masks have also been proposed to increase the sensitivity and at the same time maintain a good spatial resolution [18,19,20,21,22]. However, for nuclear medicine and imaging, where the object is normally placed close to the detector, the near-field artifacts are challenging and require, for example, mask–antimask techniques to at least partially eliminate such artifacts [23].

The foregoing proposals are highly object-dependent and a generic solution without additional technical challenges [24], or with a large FOV [25], has not been found yet. In this paper. we propose and analyze a simple, efficient and elegant generic solution to the overlapping/mutiplexing problem inherent to multi-pinhole gamma camera imaging. As in our previous study [24], we are particularly interested in obtaining a detector with unprecedented sensitivity and a spatial resolution suitable for this particular imaging application (i.e., in vivo surgical interventions), which requires a spatial resolution of the order of 0.5–1.5 cm. The solution consists of using a mobile collimator capable of either linear or rotational motion relative to the gamma-ray detector [26]. This novel approach is practically adaptable to most commercial gamma cameras systems, with resulting higher sensitivity than and comparable spatial resolution to conventional multi-pinhole imaging.

As mentioned in the Introduction, we are particularly interested in proving that this solution indeed provides a complete (non-truncated) high-quality reconstructed image without any multiplexing artifacts. We also wish to demonstrate that our proposed design has a high sensitivity; thus, for this purpose, we use sources with activities as low as 0.1–0.05 MBq and short acquisition times. The acquisition time is of the order of a few seconds for the real-time imaging procedure, and, to reduce the data processing time, small numbers of iterations and sub-iterations are used for the image reconstruction process. When the region of interest is found, a longer acquisition can be performed ≃1 min, in order to obtain a more statistically reliable (i.e., high-count) image and acquire more information about the location and depth of the source. For the evaluation of spatial resolution, however, data acquisitions up to 5 min in duration is used to obtain even more statistically reliable images.

## 2. Materials and Methods

We used GATE/GEANT4 [27,28] Monte Carlo simulations to characterize the feasibility and efficiency of the newly proposed approach to multi-pinhole gamma camera imaging. For this purpose, we consider two types of motion, linear and circular. In both cases, we consider a rectangular 100×100×3 mm${}^{3}$ Gadolinium Aluminium Gallium Garnet (GAGG) scintillator, placed at 10 mm from the center of the 3 mm tungsten collimator, as shown in Figure 1 (left). The pinholes are double-cone shaped with a semi-angular aperture of 35${}^{\circ }$; thus, the length *L*, as defined in Figure 1, is given by *L* = 10 tan (35${}^{\circ }$) ≃ 7 mm. For the gamma ray impact reconstruction in the scintillator crystal, we assume a 1 mm (FWHM) resolution in *x* and *y*, no depth of interaction resolution and a 10% energy uncertainty (FWHM).

For the linear motion, we consider the configuration shown in Figure 2 (left), where the collimator has two positions and the distance between the centers two adjacent pinholes is D=2L+d, where *d* is the inner diameter of the pinhole, which in this case is 1 mm. Therefore, we can conclude that there will be no overlapping on the detector’s surface of the corresponding incidence cones of the gamma rays, for this configuration. As presented in the next section, allowing some degree of overlapping in this configuration would generate considerable artifacts.

For the circular motion, we consider the configuration that is schematically shown in Figure 2 (right). The motion of the collimator is continuous with a rotation period of T=3.6 s. This number is chosen somewhat arbitrarily. However, mechanically speaking, it reflects the order of magnitude that is feasible for easily rotating a heavy tungsten collimator. The rotation axis is placed at 1/4 of the distance *D* between two adjacent pinholes. This way, we avoid periodicity in the solid angles that are swapped (sampled) within the FOV by the pinholes (up to 360${}^{\circ }$ rotations), which ensures that more information is retrieved from the FOV, and, thus, a better three-dimensional resolution is achieved. This would not be the case, for example, if the rotation axis were placed at exactly D/2, for which we would find equivalent pinhole configurations every 90${}^{\circ }$ rotations.

Within the circular motion subsection, two specific configurations are analyzed. First, we consider the distance *D* between two adjacent pinholes, just a in the linear case, D=2L+d, but with a larger diameter d=2 mm in order to increase the sensitivity, and L=7 mm, therefore with no overlapping. Second, to further increase the sensitivity, we consider both a large pinhole inner diameter, d=2 mm as previously, and a small degree of overlapping. Thus, with the distance *D* is given by D=2L+d−o (see Figure 1, left) where we choose o=4 mm. This corresponds to approximately 6% of overlapping between two adjacent pinholes. We observe that the noise produced by the overlapping region will not produce appreciable artifacts, due to the circular motion. This motion, as it is continuous, does not allow the artifact-related noise to *accumulate* in any well defined particular region of the FOV, but it is evenly distributed, thus mitigating the emergence of artifacts in the final reconstructed image. This, in fact, is the same phenomenon that takes place in SPECT image reconstruction, where the detector rotates about the axial symmetry axis of the region of interest, avoiding in some cases the generation of artifacts for pinholes that present some degree of overlapping. However, these overlapping effects are not easily addressed for traditional static gamma cameras.

In all three cases (linear without overlapping, circular without and with overlapping), we analyze the sensitivity, estimate the spatial resolution (in x,y and the depth z) with Derenzo-like spherical phantoms at different distances from the collimator, study the uniformity and finally compare (qualitatively) the signal-to-noise ratio for a short acquisition for a Derenzo-like phantom with 8:1 target-to-background activity ratio.

The image reconstruction method that we use is the list-mode ordered-subsets expectation maximization (LM-OSEM) algorithm [29,30,31,32,33,34,35], based on ordered sets of events (and non-indexed lines of response (LORs), as previously described in [24]). In our case, a LOR corresponds to the straight line passing through the gamma ray impact point (in the detector) and the center of the corresponding pinhole collimator. Based on the previous definition of a LOR, the reconstruction algorithm is the same as in positron emission tomography.

If we divide the whole set of LORs into L subsets and call $S_{l}$
(l=1,…,L) the *l*th subset, the image estimate $\lambda_{j}^{m,l}$ (at voxel j, *m*th iteration and *l*th subset) is given by
(1)$$\lambda_{j}^{m,l}=\frac{\lambda_{j}^{m,l-1}}{\sum_{i\in I}p_{ij}}\sum_{k\in S_{l}}p_{i_{k}j}\frac{1}{\sum_{b\in J}p_{i_{k}b}\;\lambda_{b}^{m,l-1}}\;,$$
where
(2)$$I={⋃}_{l\in L}S_{l}\;,$$
is the set of all LORs, $i_{k}$ is the LOR corresponding to the *k*th event, *J* is the set of voxels traversed by the LOR $i_{k}$ and $p_{ij}$ is the probability of an emission from voxel *j* being detected along the LOR *i*. Again, as mentioned in our previous work [24], we use an effective Monte Carlo approach to estimate the $\sum_{i\in I}p_{ij}$ matrix elements (sensitivity matrix) by simulating a high activity radioactive source that fills the whole FOV. In our case, the FOV is a 10×10×10 cm${}^{3}$ cube with 1×1×1 mm${}^{3}$ voxels. It is also worth mentioning that, for the cases in which the acquisitions are short (of the order of a few seconds) and the activity is low (of the order of 0.1 MBq), as the statistics are very low, the use of the $1/\sum_{i\in I}p_{ij}$ terms in equation (Equation 1) would result detrimental for the final reconstructed image. This is due to the fact that these matrix elements, roughly speaking, account for the geometrical sensitivity that corresponds to each voxel, and these geometrical effects are only relevant for high-count (i.e., *good-statistics*) acquisitions. Therefore, for the previously mentioned specific cases (as the statistics are going to be low), we set $\sum_{i\in I}p_{ij}=1$, ∀j∈ FOV.

For similar reasons, for the two cases corresponding to the rotational motion, we consider an *effective* set of elements $\sum_{i\in I}p_{ij}$ that contain the projections corresponding to all the positions of the pinholes during a rotation period T. As explained above, as the acquisitions are short and the activities are low, for the image reconstruction, it is better to use an effective sensitivity matrix corresponding to the sum of all contributions (and not corresponding to each instantaneous position).

## 3. Discussion

As the first part of the discussion, we focus on the sensitivity of each of the three configurations. They are given in Figure 3, evaluated along the line parallel to the *x* axis (see reference system defined in Figure 2) that passes through the center of the detector at a distance z=15 mm (left), and along the line parallel to the *z* axis also passing through the center of the detector (right). In all cases, the blue dotted line corresponds to the linear motion configuration, by considering equal contributions from both positions. The solid red curve and the dashed orange curve correspond to the continuous rotational motion, without and with overlapping.

Note that there is a factor ≃3 difference between the first and second cases, and the same is valid for the second and third cases. As we are particularly interested in significantly reducing the activity administered to the patient, the sensitivity values obtained are quite promising [4,5,6].

We also show in the following that, by reducing the activities of the analyzed phantoms, maintaining and even lowering the acquisition times, for the rotational motion with overlapping, we are able to obtain high-quality reconstructed images without overly sacrificing the spatial resolution.

### 3.1. Linear Motion

First, let us conceptually illustrate how, by introducing collimator motion, we can obtain an artifact-free (non-truncated) complete image without overlapping. As mentioned above, in both *static* gamma camera imaging and SPECT, if we do not allow at least a small amount of overlapping, the resulting reconstructed image suffers truncation (is incomplete). On the other hand, the overlapping regions might translate in many cases into serious artifacts in the final image [8,9,10,11]. Introducing a mobile collimator is effectively the same as having overlapping regions, but being able to distinguish through which pinhole a gamma ray has previously passed before reaching the overlapping zone in the detector. This is shown in Figure 4, where we can observe (left) the schematic representation of the circular basis of the allowed gamma ray incidence cones on the detector’s surface, together with the center of one pinhole for the first position of the collimator and its corresponding translation trajectory towards the second position. In Figure 4 (middle), the red dotted circles represent the detection regions corresponding to each pinhole over the detector’s surface for the second position, together with the regions corresponding to the initial position (solid black circles). One should note that, except for the borders, the non-overlapping surfaces corresponding to collimator positions 1 and 2 cover the entire detector area, which justifies that this particular solution is indeed able to solve the overlapping problem in gamma camera and SPECT imaging. All the above-presented concepts can be trivially extended to the circular motion and additionally to single photon computed emission tomography.

To reinforce the fact that, in static gamma camera imaging (without rotation about the object of study as in SPECT), and even for the mobile collimator gamma camera with linear motion, the overlapping translates into acute artifacts impossible to eliminate in the image reconstruction process, we present the following case. Consider a three-sphere Derenzo-like phantom at z=4 cm distance from the collimator, of 5 mm radius, 2 cm distance among the center of the spheres, 0.1 Mbq activity of each source and a 2 min acquisition time. By setting the distance between two adjacent pinholes to D=2L+d−o for the linear case, with L=7 mm, d=1 mm and o=4 mm, we can observe in the final reconstructed image (Figure 4, right) that, besides the three-sphere Derenzo, very pronounced artifacts are also present.

In the following, we set the *o* term to zero, meaning that we consider the non-overlapping case. We first analyze the spatial resolution in x, y, at different distances in *z* from the collimator. In Figure 5, we show the image reconstruction with two iterations and two sub-iterations of: (a) 5 mm Derenzo-like phantom, at z=4 cm, composed of 6 spheres with 5 mm diameter and 1 cm distance in between their centers, 1 min acquisition; (b) 0.1 Mbq activity and, 1 cm Derenzo-like phantom, at z=10 cm, composed of 3 spheres with 1 cm diameter and 2 cm distance in between their centers, 0.1 Mbq activity with 1 min acquisition time; and (c) 5 min acquisition time. We can observe that, the resolution in x, y lays in the range 5–10 mm for sources at 4–10 cm distance from the collimator, which is the order of magnitude that we were trying to achieve.

As for the resolution in *z*, we can conclude from the results in Figure 6 that it is in the order of 1.5 cm. We can observe that there is an (expected) image degradation for larger distances; however, one can easily visualize the two sources. Thus, the high number of holes together with the collimator motion offers fairly accurate three-dimensional information about the sources. Even if the spatial resolution remained of the same order, we would see that, for the circular motion, due to the high angular sampling, the second peak would suffer less degradation and, therefore, the reconstructed image would be more accurate.

The uniformity test is performed using a cylinder with its bases parallel to the collimator plane, placed at z= 6 cm, with 1 cm height and 5 cm diameter. The reconstructed image (five iterations and three sub-iterations), for a 3 min acquisition and 0.1 Mbq activity, is presented in Figure 7. We can observe that the image does not present any artifact and the detector exhibits excellent image uniformity in the x, y plane. Along the *z* axis, to compensate for the smaller set of projections for increasing values of *z*, we implement an exponential attenuation-like correction $I=I_{0}e^{-z}$. The corresponding profile and the *ground truth* are also shown in the same figure.

Finally, let us qualitatively analyze the signal to background ratio for a short acquisition of 4 s (2 s for each position), which would correspond real-time imaging during some surgical intervention. For this purpose, we analyze the reconstructed image (two iteration two sub-iterations) for a 1 cm Derenzo-like phantom composed of three spheres of 0.1 Mbq activity each, placed at a 5 cm distance from the collimator, and the same phantom with an additional background source. The activity of the spherical sources correspond to a 8:1 ratio activity per unit volume with respect to the background. The shape of the background source is a cube occupying the whole FOV. The image reconstruction corresponding to these two cases is shown in Figure 8, for an acquisition with no background (Figure 8a), with background (Figure 8b) and its corresponding profile (Figure 8c). In both cases, one can perfectly distinguish the three spheres. Short, several-second acquisitions are therefore sufficient for obtaining useful three-dimensional information on a volume of interest. Once this region is localized, one can perform a longer acquisition to obtain more precise information about the position, depth, dimensions, etc. of the tumors or ganglia.

### 3.2. Circular, No Overlapping

As we can conclude from the previous section, the linear configuration presents good spatial resolution ≃0.5 to 1 cm depending on the distance to the collimator. However, as we are particularly interested in drastically improving the detector sensitivity and reducing the administrated dose to the patient, for the circular design, we consider larger diameter pinholes, i.e., with d=2 mm instead of 1 mm, as considered previously. For this configuration, one can appreciate in Figure 3 that the sensitivity is 2–3 times higher than that in the previous case, and with only a slight degradation of spatial resolution. For the same Derenzo as the one shown in Figure 5a, for z=4 cm, we obtain similar results; thus, at this distance, we have no degradation. At z=10 cm, however, the obtained resolution is ≃1.5 cm (compared to 1 cm from the previous case). Note that, for a (one-)pinhole gamma camera, the approximate resolution is given by
(3)$$\begin{array}{c} R=\sqrt{{\left(\frac{\phi *(h+d)}{d}\right)}^{2}+{\left(\frac{R_{i}*h}{d}\right)}^{2}}\;, \end{array}$$
where ϕ is the diameter of the pinhole opening, *d* is the distance between the hole and the detector, *h* is the distance between the hole and the object and $R_{i}$ is the intrinsic resolution. Thus, at z=10 cm, one would expect an approximate resolution of 2.4 cm. However (similar to single photon emission tomography), due to the rotation, the final resolution is actually higher. One should also note that the image reconstruction method (MLEM) performs some de-blurring, which further improves the expected result.

The reconstructed image of the uniformity phantom is similar to that of the previous case (not shown). Let us now focus on the resolution along the *z* axis. We observe that we still have a resolution of about 1.5 cm; however, we observe a slight improvement in the reconstructed image of the second sphere, which is due to a higher angular sampling (see Figure 9).

For the target-to-background ratio, we present a 1.8 s acquisition (compared to the 4 s acquisition in the previous case) which corresponds to an acquisition up to a 180${}^{\circ }$ rotation of the collimator with respect to its initial position. This, again, represents the real-time monitoring case during some surgical intervention. The reconstructed image (two iterations two sub-iterations) for the same previous 1 cm Derenzo-like phantom composed of three spheres of 0.1 Mbq activity each, placed at a 5 cm distance from the collimator, with and without background activity, are shown in Figure 10, for an acquisition with no background (Figure 10a), with background (Figure 10b) and its corresponding profile (Figure 10c). The profile of the current acquisition is given by the black solid curve, while the dashed red line corresponds to the previous case with linear motion and 4 s acquisition.

Figure 10c shows that, as the sensitivity is higher, even if the pinhole diameter is double, one obtains better results for a shorter acquisition time. Again, this is obviously due to the higher angular sampling of the rotational motion.

In Figure 11, for comparison, to demonstrate that the reconstructed image suffers truncation and uniformity loss without the collimator motion, we show the reconstructed image for the same Derenzo as in Figure 5a and for the uniformity cylinder (Figure 7, top-left). One can easily appreciate that the image corresponding to the Derenzo-like phantom is incomplete (truncated) as there are blind regions in the FOV due to the lack of collimator motion and/or overlapping regions on the detector’s surface. For the same reason, uniformity is also lost, as can be appreciated in the same figure. If, on the other hand, we allowed some amount of overlapping without collimator motion, we would face similar artifacts such as the ones presented in Figure 4 (right).

### 3.3. Circular, with Overlapping

For the last case, we allow a small degree of overlapping, i.e., the distance between two adjacent pinholes is D=2L+d−o with *L* and *d* the same as previously and o=4 mm. This is done to further increase the sensitivity. As already mentioned, due to the small degree of overlapping and the rotational motion, this does not create appreciable artifacts. It does not affect the spatial resolution or the uniformity (the reconstructed images are very similar the ones presented already and are not shown). The real benefit can be appreciated in the real-time acquisition. In Figure 12, we present the image reconstruction for a 1.8 s acquisition without background (Figure 12a), with background (Figure 12b) and its profile (Figure 12c) given by the purple dotted line (the solid black line corresponds to the previous acquisition for circular motion and no overlapping). We observe that the signal to background ratio is similar however, the activity of each source in this case is 0.05 Mbq, which represents half of the previously employed dose. It is therefore beneficial to increase the sensitivity, even if we introduce a small overlapping percentage, as the spatial resolution is not affected and the overlapping does not create appreciable artifacts. We thus demonstrate that we can obtain high-quality results with activities as low as 0.05 Mbq.

## 4. Conclusions

In this work, we propose a novel solution to the overlapping/multiplexing problem inherent to gamma camera imaging and SPECT. Here, we only analyzed the static gamma camera imaging procedure (with a mobile collimator) and, we only focused on increasing the sensitivity rather than obtaining an outstanding image resolution. We believe that obtaining high quality images for in vivo monitoring during surgical processes, with drastically reduced radiation dose, can be highly beneficial for oncological applications, as it helps surgeons rapidly and precisely localize the region of interest, reducing the risks associated to radiation exposure for the patients and the medical staff.

Based on the previous analysis, our team will develop and probe experimentally a prototype corresponding to a gamma camera with a rotating pinhole collimator with a small amount of overlapping (the third configuration).

One could, of course, apply the previously presented ideas in another direction, focusing on enhancing the resolution, by modifying the focal distance, reducing the pinhole diameter or rotating the whole detector about the object/animal/patient of study as in SPECT. This study is left for a future publication.

## 5. Patents

Ilisie, L., Moliner, & Benlloch, J. M. Detector de rayos gamma con colimador multi-orificio y región de muestreo variable (Multi-pinhole gamma ray detector with variable sampling region), Patent application Ref. P202030173/1912005-ESP, 28/02/2020. 

## Figures and Tables

**Figure 1 sensors-21-03367-f001:**
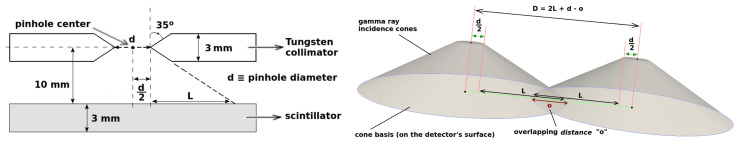
Schematic representation of the detector configuration formed by the scintillator and pinhole collimator together with different parameters used in this analysis, such as the pinhole diameter *d*, the length *L*, the distance between the detector and scintillator and their corresponding thickness, the distance *D* between two adjacent pinholes and the overlapping *distance o*.

**Figure 2 sensors-21-03367-f002:**
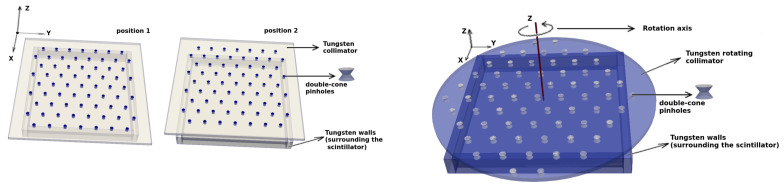
Different detector configurations: a mobile collimator with linear motion and two positions (**left**); and a circular collimator and its corresponding rotation axis (**right**). The coordinate system employed in both cases is also shown.

**Figure 3 sensors-21-03367-f003:**
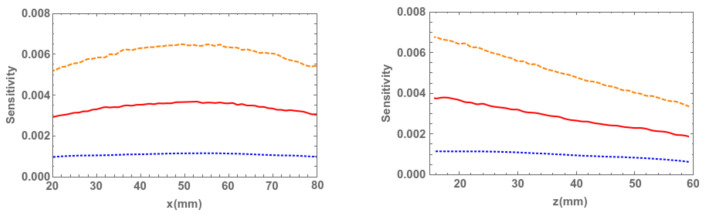
Detector sensitivity (cps/Bq) evaluated along a line parallel to the *x* axis (see reference system defined in Figure 2) passing through the center of the detector at z=15 mm, where x=50 mm represents the coordinate of the center of the detector (**left**), and along the *z* axis also passing though the center of the detector, where the *z* coordinates stand for the distance from the source to the collimator (**right**). The blue doted curve represents the linear motion, the red solid curve stands for circular motion without overlapping and orange dashed curve represents circular motion with overlapping.

**Figure 4 sensors-21-03367-f004:**
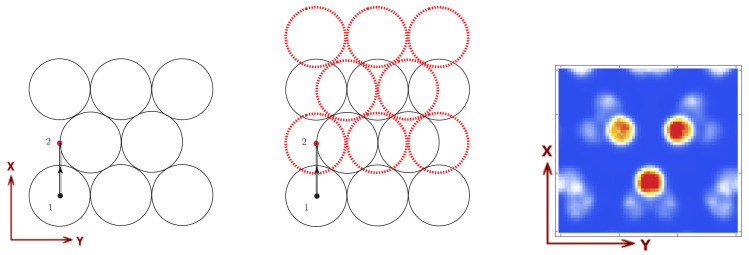
Representation of the gamma ray incidence cone bases in the non-overlapping case for the linear motion corresponding to the first collimator position (**left**) and, both the first and second collimator positions (**middle**), where the red dotted circles correspond to the second position of the collimator. Image reconstruction of a three-sphere Derenzo-like phantom for the collimator with linear motion, two-positions and overlapping (**right**) (see reference system defined in Figure 2).

**Figure 5 sensors-21-03367-f005:**
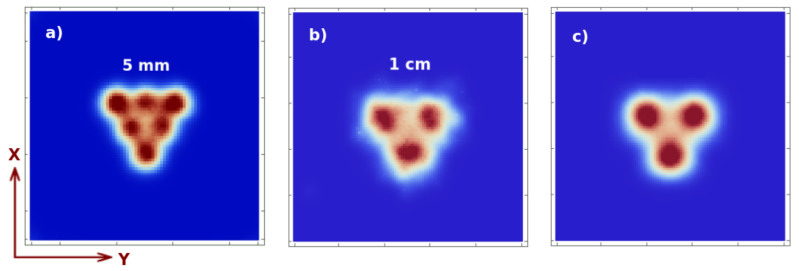
Image reconstruction of 5 mm Derenzo-like spherical phantom (with 0.1 MBq activity of each sphere) at z=4 cm (see reference system defined in Figure 2) and 1 min acquisition time (**a**); 1 cm Derenzo at z=10 cm with 1 min acquisition time (**b**); and 5 min acquisition time (**c**).

**Figure 6 sensors-21-03367-f006:**
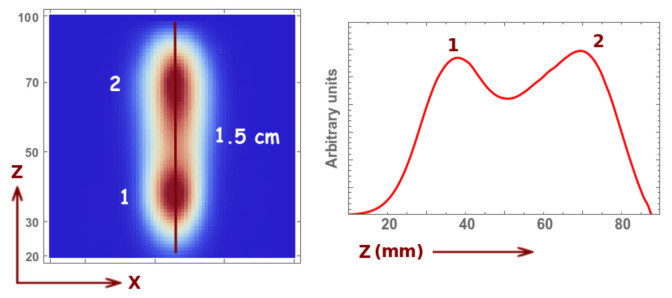
Image reconstruction (five iterations and three sub-iterations) and profile for a two-sphere phantom with 1.5 cm diameter and 3.0 cm distance in between their centers, 1 min acquisition time and 0.1 Mbq activity (of each sphere). The distance in *z* stands for the distance from the collimator (see reference system defined in Figure 2).

**Figure 7 sensors-21-03367-f007:**
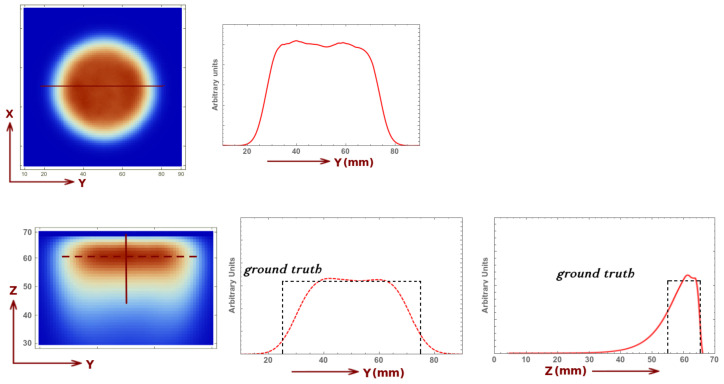
Image reconstruction of the uniformity phantom and its corresponding profile in the x,y plane (**top**) and the y,z plane (**bottom**). The *ground truth* profile is also shown for the last case (see reference system defined in Figure 2).

**Figure 8 sensors-21-03367-f008:**
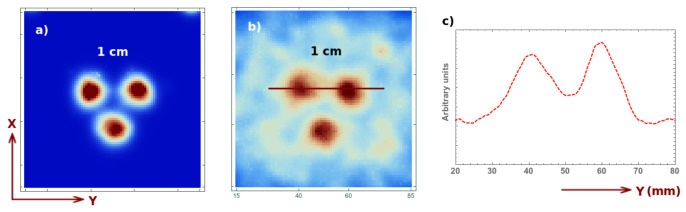
Image reconstruction of a 4 s acquisition of a 1 cm Derenzo at 5 cm from the collimator without background (**a**), with background (**b**) (8:1 signal to background ratio) and its profile (**c**) (see reference system defined in Figure 2).

**Figure 9 sensors-21-03367-f009:**
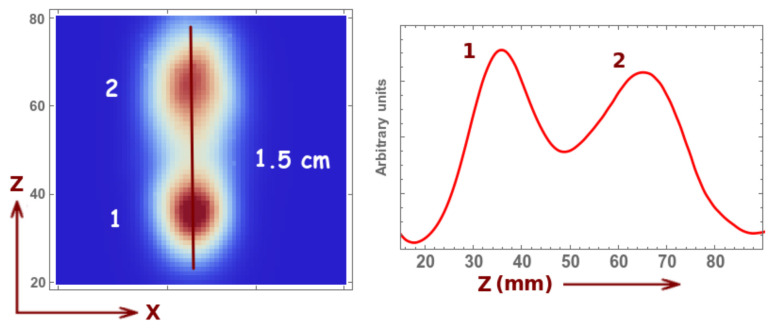
Image reconstruction (five iterations and three sub-iterations) and profile for a two-sphere phantom with 1.5 cm diameter and 3.0 cm distance in between their centers, 1 min acquisition time and 0.1 Mbq activity (of each sphere). The distance in *z* stands for the distance from the collimator.

**Figure 10 sensors-21-03367-f010:**
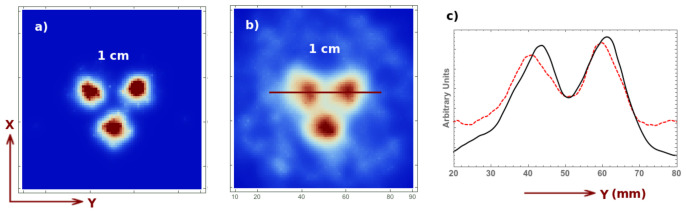
Image reconstruction for a 1.8 s acquisition of a 1 cm Derenzo and 0.1 Mbq activity for each sphere, at 5 cm from the collimator, without background (**a**), with background (**b**) (8:1 signal to background ratio) and its profile (**c**) for the current acquisition (solid black curve) and for the previous acquisition (red dashed curve).

**Figure 11 sensors-21-03367-f011:**
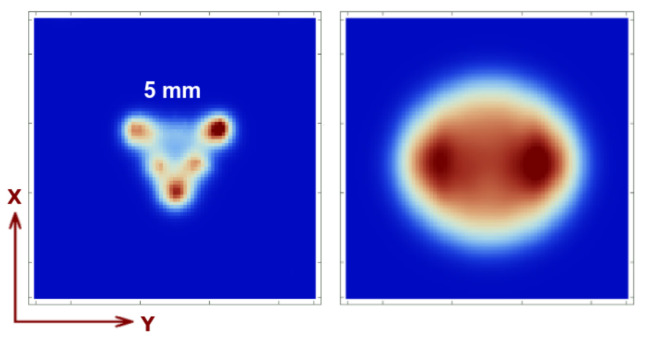
Image reconstruction of 5 mm Derenzo like spherical phantom (with 0.1 MBq activity of each sphere) at z=4 cm and 1 min acquisition time (**left**) and the uniformity phantom (**right**) placed at z=5 cm with 1 cm height and 5 cm diameter, for a 3 min acquisition and 0.1 Mbq activity.

**Figure 12 sensors-21-03367-f012:**
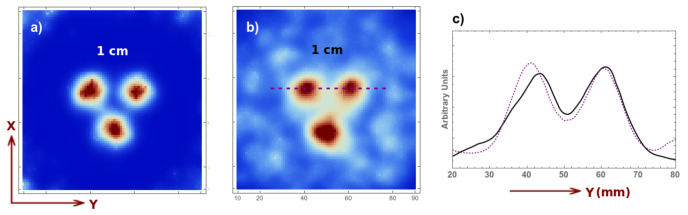
Image reconstruction for a 1.8 s acquisition of a 1 cm Derenzo and 0.05 Mbq activity for each sphere, at 5 cm from the collimator without background (**a**), with background (**b**) (8:1 signal to background ratio) and its profile (**c**) for the current acquisition (dotted purple curve) and for the previous acquisition (solid black curve).

## Data Availability

The data presented in this study are available on request from the corresponding author.

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
