# Peer review of "Gamma Camera Imaging with Rotating Multi-Pinhole Collimator. A Monte Carlo Feasibility Study"

_sensors, 2021, doi:10.3390/s21103367_

Round 1

Reviewer 1 Report

Review report for article ID sensors-1208320 titled “Gamma camera imaging with rotating multi-pinhole collimator. A Monte Carlo feasibility study”

General evaluation:

This is an interesting paper presenting a novel approach for gamma camera imaging using a relative motion between the collimator and the gamma-ray detector. This approach has been patented. It could be implemented into the available camera devices, and the feasibility of such implementation has been tested in three possible designs with the Monte Carlo simulations. It is shown that this approach allows reducing the employed dose by a factor of at least 10, operating in a safer radiation environment. This is a very promising achievement, and I believe a reader will be impressed by other results presented in the scope of this article. For a better understanding of its content, I provided nine comments and questions to the authors listed below. All comments should be answers in the “Response to the Reviewer” file, particular attention please devote to my comments 7-9.

Comments to authors:

  • Page 1, line 11: Please correct English in the phrase: “We use Monte Carlo simulations in order the assess …” It should be changed to “We use Monte Carlo simulations in order to assess …”
  • Page 1, line 16: Please confirm that the 2.0 cm spatial resolution in the detection of the gamma source depth is precise enough for the aim on which the planar gamma camera is used. It seems that such resolution is too coarse because the planar camera could provide a better resolution. What are the acquisition time and source activity (is it ~0.1 MBq or ~1 MBq) for the image reconstruction process with the 2-cm resolution?
  • Page 2, lines 36-37: Please add a word “the reduction” in the ending phrase “and thus of the relatively high cost currently associated with in-vivo molecular imaging procedures.” So the correct phrase will be “and thus the reduction of the relatively high cost currently associated with in-vivo molecular imaging procedures.”
  • Page 3. Why you mentioned “as shown in Figure 1 (left)”? There are no left or right sides in Fig. 1.
  • Page 3, line 130: Again, incorrect reference to “Figure 1, left”
  • Page 6, line 221: Please correct “tree” to “three” in the phrase “tree-sphere Derenzo-like”.
  • Figure 6: Scale in the right-side picture should be mm instead of cm because is corresponds to a profile of profile for a two-sphere phantom with 2 cm diameter, distance between peaks 1 and 2 is ~2 cm.
  • In all the rest Figures 7-10, and Fig. 12 the units of the scales should be in (mm) instead of (cm).
  • Page 8, lines 250-251: I can advise how to solve the observed in Fig. 7 problem of “typical elongation due to the very small set of projections that we are using for the image reconstruction”. You should take into account the gamma-flux attenuation in z-direction. The attenuation follows the exponential law with a linear coefficient of attenuation. This law and coefficients are available in many textbooks of radiation shielding. Using the gamma attenuation law, the shape of image reconstruction will correspond to the actual shape of the source, even with small number of sampled particles. The reconstructed shape will be reproduced correctly, saving time of gamma irradiation. If you need more details how to implement this idea in practice, please let me know in your response to the reviewers.

Reviewer 2 Report

The paper deals with an interesting topic imaging in surgical tasks. In the present version the paper needs improvements. Starting from the introduction authors should provide a wider picture on the research domain in order to better conestualise their work. Some more references should be added as for instance the following ones:

Tanzi, L., et al. (2020). Intraoperative surgery room management: a deep learning perspective. The International Journal of Medical Robotics and Computer Assisted Surgery16(5), 1-12.

LEIBETSEDER, Andreas, et al. Image-based smoke detection in laparoscopic videos. In: Computer assisted and robotic endoscopy and clinical image-based procedures. Springer, Cham, 2017. p. 70-87.

Regarding the methodlogical section authors should provide a more clear picture of the overall method for giung further in the more specific methodiological stages. Also the experimental validation should be improved by proviging data regarding usages specifi scenarios and a more systemic performances data set. Also the experimental setting should better described

Round 2

Reviewer 1 Report

Thank you for addressing, point by point, all my nine comments, adequate answering, and successful implementation in the revised version. I recommend acceptance of your revised manuscript.

Reviewer 2 Report

Authors have improved the scientific level of the paper